# A Study of the Impact of COVID-19 on Urban Contact Networks in China Based on Population Flows

**Xuejie Zhang [1,2], Jinli Zhao [1,2], Haimeng Liu [3,4], Yi Miao [1,2], Mengcheng Li [1,2] and Chengxin Wang [1,2,*]**

1   College of Geography and Environment, Shandong Normal University, Jinan 250358, China
2   Collaborative Innovation Center of Human-Nature and Green Development, Universities of Shandong, Jinan 250358, China
3   Institute of Geographic Sciences and Natural Resources Research, Chinese Academy of Sciences, Beijing 100101, China
4   College of Resources and Environment, University of Chinese Academy of Sciences, Beijing 100049, China
*   Correspondence: 110105@sdnu.edu.cn

**Abstract:** The emergence and enduring diffusion of COVID-19 has had a dramatic impact on cities worldwide. The scientific aim of this study was to introduce geospatial thinking to research related to infectious diseases, while the practical aim was to explore the impact on population movements and urban linkages in the longer term following a pandemic outbreak. Therefore, this study took 366 cities in China as the research subjects while exploring the relationship between urban contact and the outbreak of the pandemic from both national and regional perspectives using social network analysis (SNA), Pearson correlation analysis and multi-scale geographically weighted regression (MGWR) modeling. The results revealed that the number of COVID-19 infections in China fluctuated with strain variation over the study period; the urban contact network exhibited a significant trend of recovery. The pandemic had a hindering effect on national urban contact, and this effect weakened progressively. Meanwhile, the effect exhibited significant spatial heterogeneity, with a weakening effect in the eastern region ≈ northeast region > central region > western region, indicating a decreasing phenomenon from coastal to inland areas. Moreover, the four major economic regions in China featured border barrier effects, whereby urban contact networks constituted by cross-regional flows were more sensitive to the development of the pandemic. The geostatistical approach adopted in this study related to infectious disease and urban linkages can be used in other regions, and its findings provide a reference for China and other countries around the world to respond to major public health events.

**Keywords:** COVID-19; urban contact network; population flows; multi-scale geographically weighted regression model; China

## 1. Introduction

An epidemic refers to infectious diseases, which are capable of spreading infection between populations and animals. In accordance with the classification of epidemics by the World Health Organization [1], the COVID-19 outbreak in 2019, as a novel infectious disease transmitted predominantly by respiratory droplets and contact, falls into the highest category of "pandemic" [2]. Meanwhile, it has also been evaluated by the United Nations as the greatest challenge facing the entire human race since World War II, which has prompted a surge in research by professional scholars from various disciplines [3,4].

The results of numerous analyses of the externality of the development of the pandemic and exploration of the impact mechanisms demonstrate that the outbreak of the COVID-19 pandemic has brought about a tremendous impact on the development of cities and people's lives throughout the world while underlining the intimate and complicated relationship between population flows, urban contact and the pandemic [5–8].

The inter-city population flows stand out as one of the most significant forms of spatial mobility, which embodies the complicated connections among numerous elements in the territorial system of the human–land relationship [9]. Researchers have employed various theories, models, methods and techniques to probe the characteristics and patterns of urban contact from the perspective of population flows [10–12], among which the construction of urban contact networks on the basis of the gravitational model has been extensively recognized [13,14]. Nevertheless, as a result of coarse statistical granularity, this method fails to represent the inter-city population flows in a veritable and accurate manner. Over recent years, objective records, such as sensor technology, satellite positioning, wireless communication and mobile internet, have introduced breakthroughs in research, with massive amounts of daily population flow data facilitating the quantitative analysis of population flows within urban contacts. There have been numerous related studies, which have mapped urban contact networks using Tencent and Baidu migration data in order to analyze the structural characteristics of population flow networks and their influencing factors during sudden public health events [15,16]. These research results have concentrated on the analysis of the impact of population flows on the development of the pandemic, as well as the opportunities, challenges and countermeasures for urban development in the context of the pandemic [17,18]. It is commonly acknowledged that cities with high population flow dynamics and high density have found it more difficult to contain the pandemic [19–21]. The effect of influence attributed to population structure, occupational composition and density has differed significantly, with traders contributing significantly more to transmission and promotion in comparison to the rest of the population [22,23]. In the meantime, psychological changes during the "pandemic" have also differed significantly across flow situations [24,25], with spatial heterogeneity in the hierarchical decline of urban hierarchies as a result of the pandemic [26]. In the context of the pandemic, controlling inter-city population flows exerted a substantial preventive and control role in the initial phase of the pandemic [27], and the comparatively high level of digitalization boosted the capability of cities to cope with shocks and their resilience [28], while the application of a coping strategy scoring tool elevated the scientific nature of the coping strategies [29]. Methods such as ordinary least squares regression, logistic regression and stepwise regression are commonly employed to study the relationship between urban contact and the pandemic; however, these methods have neglected the spatial impact of epidemic transmission to a large extent [30]. In contrast, geographically weighted regression and its optimized upgraded series of models (GTWR, GWPR, MGWR, GLM, etc.) have made improvements in the area of spatial heterogeneity by concentrating on the local effects of objects [31–33]. The application of this category of models has pioneered a general analytical paradigm for the prevention and control of a pandemic, analysis of disease transmission pathways and the responses to sudden public health events.

Based on the literature review above, we found that many research studies have focused on the impact of pandemic outbreaks during an initial short period of time [34–36], while social science research over a longer period of time is scarce due to the low availability and high complexity of handling ongoing real epidemic data. Therefore, to fill the current research gap, this study analyzed the impact of the epidemic on urban linkages over a two-year period in 366 cities in China, in an attempt to address the following three scientific questions: "Are population movements and urban contact still affected by the epidemic for a prolonged period after the outbreak? How are they affected? Are the effects consistent across regions?".

For the sake of tackling these issues, this study took 366 cities in China as the research subjects to compile the variations in the number of patients diagnosed with COVID-19 in the two years from 1 July 2020 to 30 June 2022. This study employed Ucinet and Gephi to map and calculate the urban contact network at multiple scales based on the perspective of flow space while applying the Pearson correlation analysis and multi-scale geographically weighted regression (MGWR) models to probe the overall correlation and spatial heterogeneity of the relationship between urban contact and the epidemic from both

national and regional perspectives (Figure 1). Meanwhile, this study sought to uncover the spatial and temporal characteristics of the impact of the current pandemic on urban contact, as well as to present a reference research paradigm for future studies from a flow perspective, thereby facilitating the formulation of more scientific public health interventions.

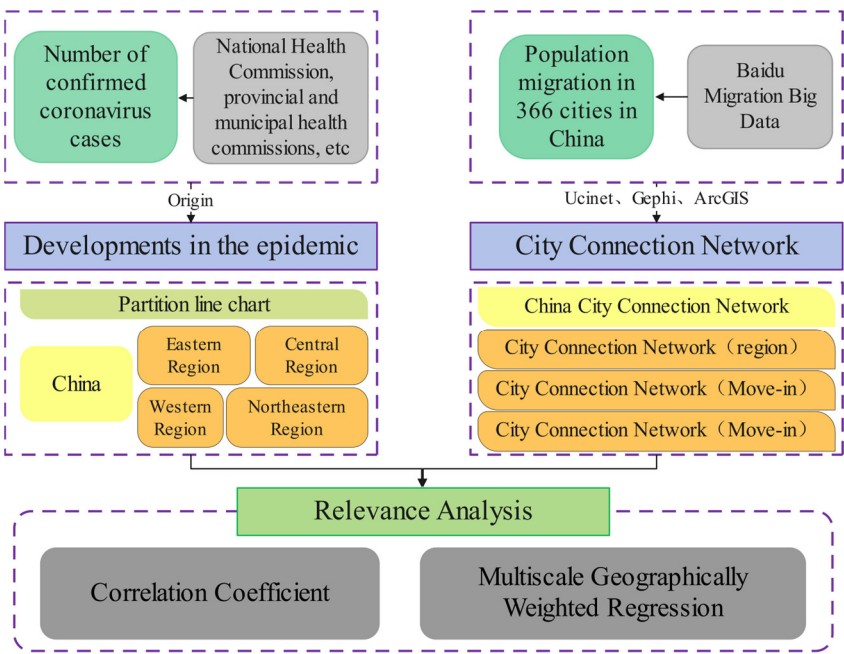

**Figure 1.** Research roadmap.

## 2. Theoretical Analysis and Research Hypothesis

### 2.1. Existing Mechanisms for the Influencing Role of the Development of the Pandemic in Urban Contact

Numerous available studies have probed the relationship between "population flows" and epidemic transmission in urban contact, identifying this influencing role to be complicated and interactive in both directions [37]. Under theoretical conditions, if an infected person appears in a city, without restricting population flows, the number of infected people will increase dramatically over time as a result of intra-city population flows, whereas cross-city population flows will carry the virus to other cities and contribute to the migration and spread of the virus (Figure 2). As a consequence, population flows are perceived to be a significant factor influencing the inter-city epidemic transmission model.

Nevertheless, the actual situation is more complicated than the theoretical one. In the case of COVID-19, for instance, during the period following the emergence of diagnosed cases in Wuhan, China, the spread of information was significantly faster than the spread of population contact. Following the explicit emergence of the "person-to-person" phenomenon, population flows were constrained at both the policy and individual awareness levels, thereby allowing the pandemic to be brought under control [38,39]. Given the situation of normalized pandemic preparedness, the cities where the infected appeared placed more emphasis on small-scale silence within the community or even at the household level. With the resumption of cross-city population flows, the variation in the intensity of direct and indirect impacts of the development of the pandemic on "population flows" in urban contacts can, to a certain extent, demonstrate the recovery and resilience of the cities. It is possible to define the phenomenon where urban contacts are still highly sensitive to the development of the pandemic as urban post-pandemic stress disorder and refer to psychological concepts.

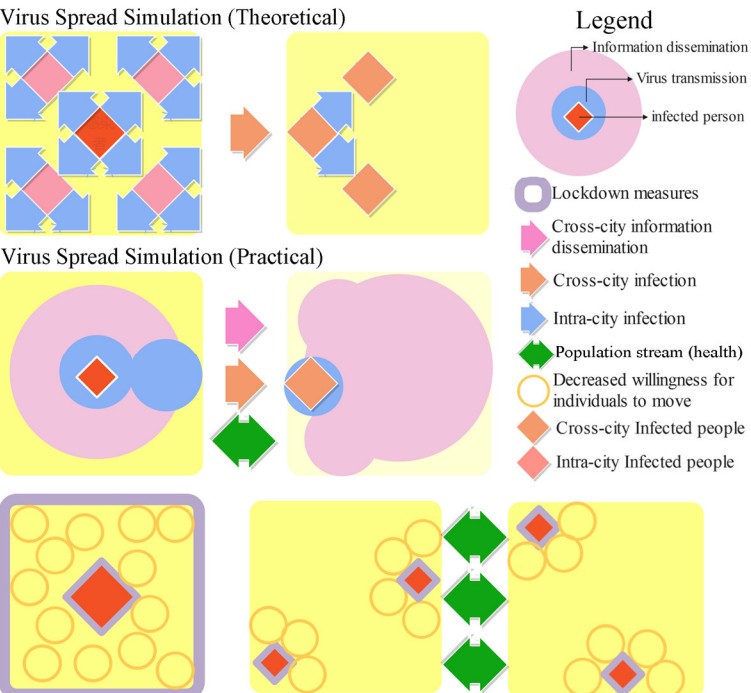

**Figure 2.** Virus diffusion simulation.

The available studies have mainly concentrated on the contribution of urban contact to the development of the pandemic at the very beginning, whereas the interaction between the two cannot be disregarded, especially with regard to the variation in the sensitivity of urban contact to the pandemic over a prolonged period of time. Accordingly, the variation over time in the influencing role of the pandemic in urban contacts in the context of population flows deserves to be probed in depth, thereby assisting future regional administrators in their decision making with regard to local adaptation and to adroitly guide actions according to circumstances, as well as to guarantee precise and scientific pandemic prevention.

**Hypothesis 1.** *For a prolonged period of the pandemic outbreak, urban contacts based on the spatial perspective of population flows are still negatively influenced by the pandemic, while such a negative influencing role has been diminishing.*

In the empirical evidence, the correlation coefficients and regression coefficients will be calculated globally and locally using the Pearson correlation analysis and the MGWR model in order to test the hypothesis.

### 2.2. Spatial Differences in the Influencing Role of the Development of the Pandemic in Urban Contact

The second law of geography, also referred to as the law of spatial heterogeneity, encompasses the differences between surface features resulting from spatial segregation [40]. The presence of spatial heterogeneity affects the locational choices of subjects, which in turn leads to some cities having high human traffic and intense inter-city connections, while some cities have low human traffic and marginal positions in the contact network. The holistic trend of the whole will overlook the peculiarities of local areas, with the obtained results being homogenized, while the holistic level of the overall analytical treatment of data of all variables will inevitably conceal some real correlations at the local level. Heterogeneity enables the condensation of flows and relationships to proceed through various processes and exhibits distinct forms, making spatial heterogeneity still a consideration in the influencing role.

**Hypothesis 2.** *There is significant spatial heterogeneity in the sensitivity of urban contacts to the development of the pandemic, with substantial variation in the negative influencing role of the pandemic within networks of varying scales (especially intra-regional urban contact networks).*

As a consequence, this study obtained daily data on the number of people diagnosed in 366 cities nationwide to characterize the development of the pandemic in cities while employing daily data on population flows to construct spatial networks of population flows from various regional and national scales to characterize urban contacts. Furthermore, the Pearson correlation analysis and the MGWR method were adopted to analyze the influencing role of the pandemic outbreak over a prolonged period of time in both global and local dimensions to verify the above hypotheses.

## 3. Materials and Methods

### 3.1. Data Sources

In this study, city-scale data on the new diagnosed cases of COVID-19, namely cumulative diagnosed cases, were employed to characterize the development of the urban pandemic. The public data from the national, provincial and municipal health care commissions from 1 July 2020 to 30 June 2022 were crawled with Python, while the data were collected, cleaned and screened for outliers by taking prefecture-level cities, municipalities and autonomous regions as the research unit.

Moreover, the Baidu Migration Big Data from the Baidu Migration Platform "https://qianxi.baidu.com (accessed on 2 July 2022)" were utilized to characterize urban contacts from a spatial perspective of population flows. As one of the most extensive technical service platforms for LBS data in China, the Location-Based Service (LBS) Open Platform of Baidu Maps (http://lbsyun.baidu.com/, accessed on 1 July 2022) uses GPS, IP addresses, mobile base stations, WiFi and hybrid positioning to collect users' locations at different points in time and scientifically determine whether they have migrated or not [41]. The Baidu Migration Platform is able to display comprehensive, real-time population migration data obtained using Baidu's cloud computing platform. The timing was precise to the daily population flows of 366 cities from 1 July 2020 to 30 June 2022, with five fields marked as "Inflow", "Time", "Departure City", "Arrival City", etc. The data feature real-time and high coverage, enabling the demonstration of the trajectory and characteristics of population flows within the study period in a full, dynamic, immediate and intuitive manner in order to intuitively grasp the short-term population flow situation among cities, as well as the intensity of inter-city connections.

The data required for the influencing factor index system were obtained from the China Ecological and Environmental Status Bulletin and the socioeconomic statistics bulletin of each city, in addition to the CSMAR Economic and Financial Research Database, Landsat remote-sensing data (30 m accuracy), DMSP/OLS night-time lighting data and NPP/VIIRS night-time lighting data. Meanwhile, this study also conducted normalized pre-processing, the variance inflation factor (VIF) test and the lag test on the data in order to achieve scientific and reliable research results.

### 3.2. Study Area

The COVID-19 outbreak in late 2019 was a significant shock to the global economy, in which China was one of the largest and most effective contributors to global pandemic preparedness [42]. This study concerned 366 cities in China (4 municipalities directly under central government jurisdiction, 333 prefecture-level cities/autonomous prefectures/regions/leagues and 29 province-administered county-level cities) with high representativeness and reference to the impact of the pandemic on urban contact two years after the normalized management of the pandemic. Given the availability of data, the study excluded the Taiwan Province, the Hong Kong Special Administrative Region and the Macao Special Administrative Region. In December 2019, multiple cases of COVID-19

were diagnosed in Wuhan, China, with the city of Wuhan beginning to be sequestered in January 2020. In May of the same year, China entered a phase of normalized management for the prevention and control of COVID-19, where cases imported from abroad were fundamentally controlled, while the positive and progressive trend of the local pandemic situation in China continued to be consolidated.

In view of the regional differences in socioeconomic development in China, this study constructed networks in accordance with four major economic regions, namely the eastern region, central region, western region and northeastern region, in order to probe the variations in the influencing role at different scales (Table 1).

**Table 1.** Division of four major economic regions in China.

| Region | Provinces Included |
|---|---|
| Eastern Region | Beijing, Tianjin, Hubei Province, Shanghai, Jiangsu Province, Zhejiang Province, Fujian Province, Shandong Province, Guangdong Province, Hainan Province |
| Central Region | Shanxi Province, Anhui Province, Jiangxi Province, Henan Province, Hubei Province, Hunan Province |
| Western Region | Inner Mongolia Autonomous Region, Guangxi Zhuang Autonomous Region, Chongqing, Sichuan Province, Guizhou Province, Yunnan Province, Tibet Autonomous Region, Shaanxi Province, Gansu Province, Qinghai Province, Ningxia Hui Autonomous Region, Xinjiang Uygur Autonomous Region |
| Northeastern Region | Liaoning Province, Jilin Province, Heilongjiang Province |

*3.3. Methodology*

3.3.1. Social Network Analysis

Social network analysis (SNA) stands out as the most commonly used network analysis tool to build networks and analyze the spatial structure by taking advantage of the population flows between cities. The connections between cities have demonstrated the spatial network structure under the joint effects of "central place" and "flow space" [12].

Network density is a representation of the structural tightness between the nodes in the network, which is embodied in the ratio of the actual associated edges to the theoretical maximum number of possible associated edges in the network. The larger the value, the smoother, faster and stronger the flows of elements between urban networks. The formula is as follows:

$$D = \sum_{i=1}^{k} \sum_{j=1}^{k} d\left(n_i, n_j\right) / k\left(k-1\right) \tag{1}$$

where $D$ is the network density, $k$ is the number of urban nodes, and $d$ is the actual associated edge existing between two actual points, and $n_i, n_j$ represent the actual point in cities $i$ and $j$.

With regard to the characteristics of the network nodes, the degree centrality indicates the rank of this node in the network, which is a structural location indicator, where the higher the value, the greater the influence of the node. The calculation formula is as follows:

$$C_{AD_i} = \sum_{j=1, \; i \neq j}^{n} G_{ij} \qquad G_{ij} = M_i * R_{ij} \tag{2}$$

where $C_{AD_i}$ denotes the degree centrality of city $i$; $G_{ij}$ denotes the degree of correlation of population flows from city $i$ to city $j$; the size index of emigration from a city,

denoted as $M_i$, is available on the Baidu migration platform for horizontal comparison between cities; and $R_{ij}$ represents the proportion of the size of the population emigrating from city $i$ to city $j$ to the total number of emigrants from city $i$.

This study constructed an urban contact network from the perspective of population flows through SNA, where the network density and degree centrality characteristics were explored.

### 3.3.2. MGWR

The classical OLS is a simple linear regression, but it does not take into account differences in spatial variables due to the non-stationarity of spatial locations. Geographically weighted regression (GWR) refers to a local regression model on the basis of constructing a spatial weight matrix, which takes into account spatial non-stationarity while being commonly applied in the analysis of influencing factors in space. However, the network of urban links in the perspective of population flows is not only influenced by the development of the epidemic but also by natural, economic and political factors. When multiple explanatory variables are present, the actual scale of action of the different explanatory variables is not consistent. Nevertheless, multi-scale geographically weighted regression (MGWR) loosens the assumption that all processes to be modeled are at an identical spatial scale, which is capable of satisfying the issue of smoothing the variables at their respective spatial levels, using a specific bandwidth for each variable as an indicator of the role of each spatial process, as well as employing a multi-bandwidth approach to produce a spatial process model that is closer to reality and, therefore, usable. The formulation is expressed as [43]

$$y_i = \sum_{j=i}^{k} \beta_{bjw}(u_i, v_i)x_{ij} + \varepsilon_i \tag{3}$$

where $y_i$ denotes the dependent variable of the $i$ element, $x_{ij}$ denotes the attribute value of the independent variable $j$ in position $i$, $\beta_{bjw}$ denotes the bandwidth used for the regression coefficient of the $j$ variable, $(u_i, v_i)$ serve as the spatial coordinates of the $i$ element, and $\varepsilon_i$ is the residual.

$$\hat{\varepsilon} = y - \sum_{j=1}^{k} \hat{f}_j \tag{4}$$

MGWR sets the initial state with the classical geographically weighted regression (GWR) estimation, followed by calculation of the initialized residual $\hat{\varepsilon}$, with iterative calculations to search for the optimal bandwidth.

In Equation (4), the initial residual $\hat{\varepsilon}$, additive term $\hat{f}_j$ and independent variable $x_i$ perform GWR weighting on $i = 1, \cdots, n$ to uncover the optimal $\beta_{bjw}$. Afterward, the computation is continuously iterated until one complete iteration is finished where all variables $k$ are traversed. This cycle is iterated in such a manner and repeated until the estimation converges to the convergence criterion.

## 4. Results

### 4.1. Spatial and Temporal Distribution Characteristics of the Development of COVID-19

As shown in Figure 3, the number of diagnosed cases in China from July 2020 to December 2021 was comparatively low and stable at less than 3000, while the number increased dramatically from January 2022 and peaked at more than 25,000 in April 2022. The year when the pandemic started was 2020, and China entered pandemic normalization in July 2020 under the non-pharmaceutical intervention of the state. Over the course of maintaining a stable pandemic, the number of diagnosed cases increased marginally in January and February of each year, which is the time of the Chinese New Year, when the homecoming fever provided an avenue for the spread of the virus. Nevertheless, from August to December 2021, the Delta mutant strain of COVID-19 was transmitted to China with even stronger infectivity, resulting in a localized rebound of the pandemic. The emergence of the Omicron mutant strain led to a surge in numbers in 2022 in comparison to the previous year, whereby the Omicron mutant strain was more infectious with a shorter incubation period, which contributed to a massive outbreak in China and a surge in numbers. In the meantime, the warm weather—with rising temperatures and heavy humidity being conducive to the survival and breeding of bacteria, viruses and other micro-organisms—presented a favorable opportunity for the occurrence of COVID-19. Nevertheless, following effective control by the state, the increase in the number of people diagnosed started to decline in May and June, with a basic return to the average in June 2022.

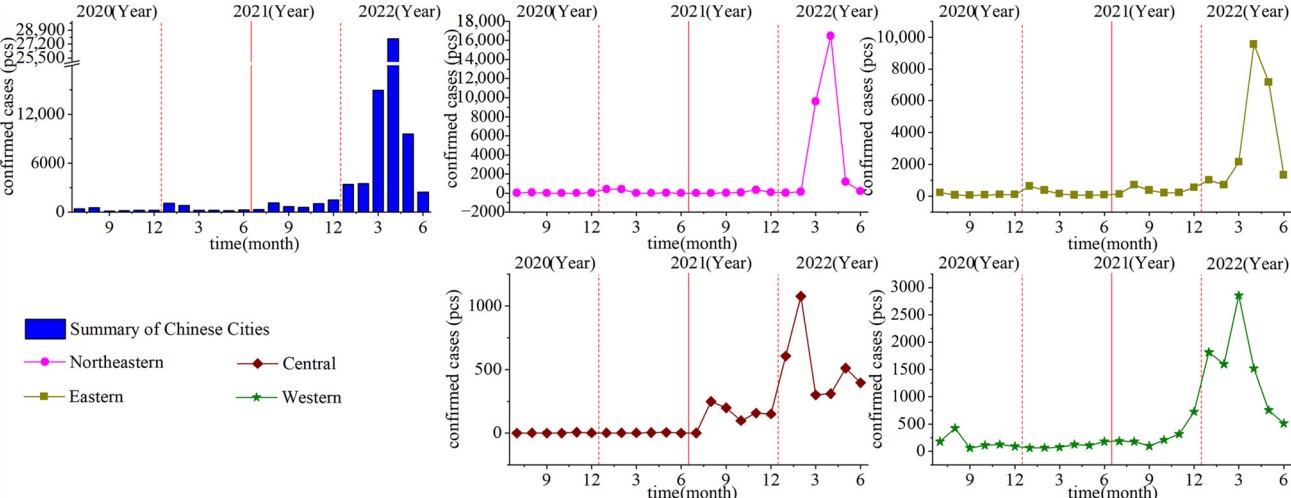

**Figure 3.** Graphs of the number of diagnosed cases of COVID-19.

The trend of COVID-19 in the northeastern region is substantially analogous to that of the eastern region, with stable control from June 2020 to February 2022. There was only a slight increase in individual months, which was identical to the national pandemic; however, there was a surge in numbers after February 2022, with outbreaks in Jilin and Shandong. Henan Province in the central region experienced outbreaks at the time of the emergence of two mutant strains, resulting in an increase in the number of people diagnosed with COVID-19. The western region was in higher alignment with the aggregate situation in China.

### 4.2. Characteristics of Urban Contact Networks from the Perspective of Population Flows

The population flows between cities formed urban contact networks. With a view to mirroring the real characteristics of urban contact networks, this study used the Ucinet software to characterize the population flow between two cities (with arrows pointing to the inflow, and the thickness of the line representing the intensity of the flow) and create the Chinese urban linkage network from the perspective of population flow (Figure 4). In

addition, the study calculated the density of urban contact networks at various scales (Table 2).

**Table 2.** Network density nationwide and in each of the four regions.

| Time | All | Eastern Region | | | Central Region | | | Western Region | | | Northeastern Region | | |
|---|---|---|---|---|---|---|---|---|---|---|---|---|---|
| | | In-House Network | Move-In Network | Move-Out Network | In-House Network | Move-In Network | Move-Out Network | In-House Network | Move-In Network | Move-Out Network | In-House Network | Move-In Network | Move-Out Network |
| 2020 3rd | 0.203 | 1.039 | 0.015 | 0.015 | 0.599 | 0.011 | 0.011 | 0.295 | 0.007 | 0.007 | 1.007 | 0.002 | 0.002 |
| 2020 4th | 0.197 | 1.030 | 0.015 | 0.013 | 0.643 | 0.010 | 0.011 | 0.263 | 0.006 | 0.007 | 0.947 | 0.002 | 0.002 |
| 2021 1st | 0.194 | 0.920 | 0.015 | 0.015 | 0.693 | 0.012 | 0.011 | 0.289 | 0.006 | 0.007 | 0.693 | 0.001 | 0.001 |
| 2021 2nd | 0.244 | 1.256 | 0.017 | 0.018 | 0.809 | 0.015 | 0.013 | 0.332 | 0.007 | 0.008 | 1.166 | 0.002 | 0.002 |
| 2021 3rd | 0.223 | 1.158 | 0.016 | 0.016 | 0.671 | 0.013 | 0.012 | 0.319 | 0.007 | 0.007 | 1.148 | 0.002 | 0.002 |
| 2021 4th | 0.243 | 1.359 | 0.015 | 0.015 | 0.847 | 0.013 | 0.011 | 0.317 | 0.006 | 0.006 | 0.959 | 0.001 | 0.002 |
| 2022 1st | 0.241 | 1.117 | 0.019 | 0.017 | 0.868 | 0.014 | 0.014 | 0.369 | 0.007 | 0.008 | 1.072 | 0.001 | 0.002 |
| 2022 2nd | 0.196 | 1.005 | 0.011 | 0.010 | 0.664 | 0.009 | 0.009 | 0.327 | 0.005 | 0.005 | 0.727 | 0.001 | 0.001 |

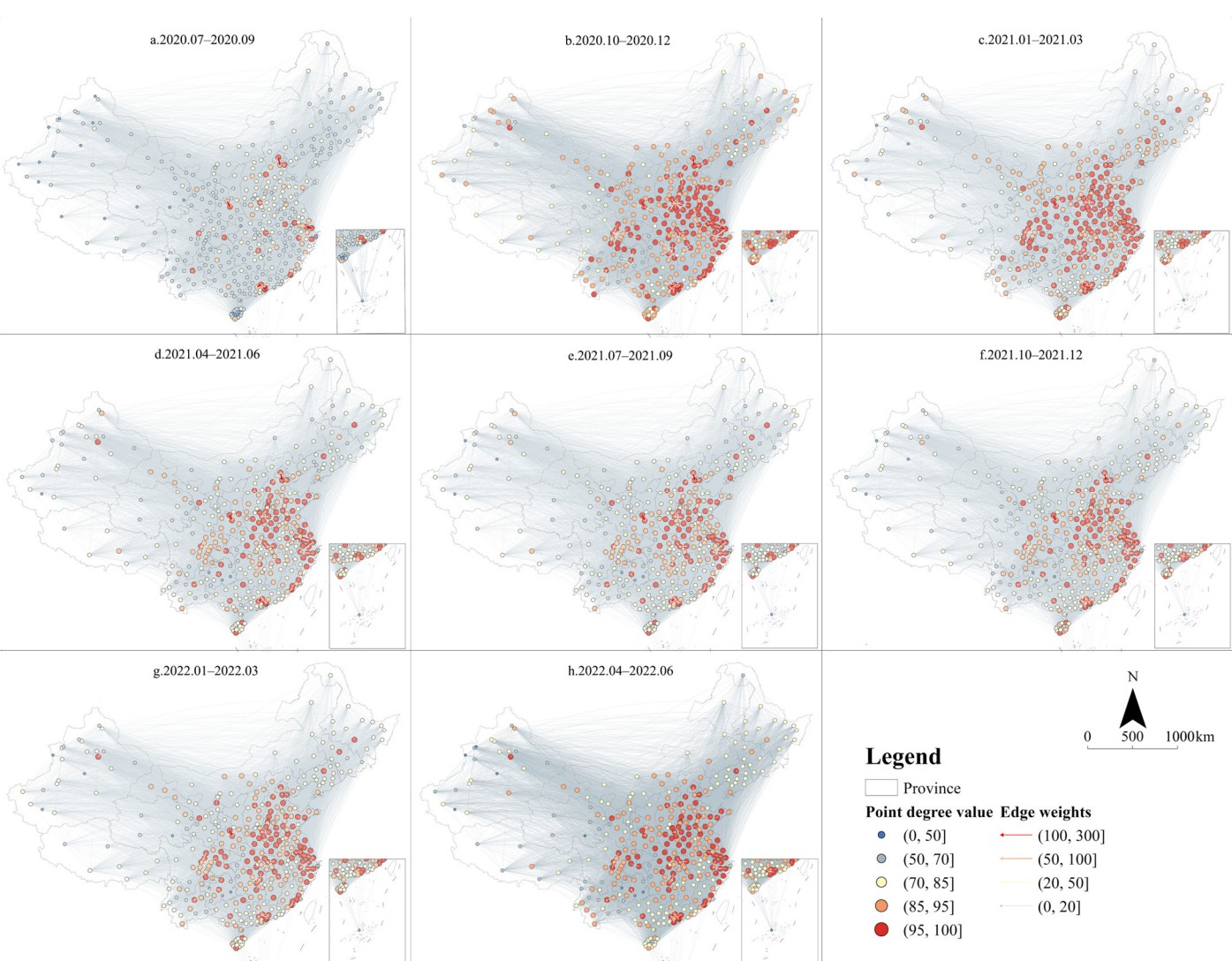

**Figure 4.** Urban contact networks from the spatial perspective of Chinese population flows.

In light of Figure 4, it can be seen that over time, the urban centrality high-value areas show an expansion, experiencing gradual extension from individual national central cities and provincial capitals in the eastern and central regions in the third quarter of 2020 to the west, into a patchy distribution behind. The overall recovery of urban contact was

significant within the two years, with the central and eastern regions prioritizing the start of population flows, resulting in a high degree and wide range of relative urban rank improvement, followed by the western and northeastern regions recovering in succession while still remaining at a lower rank in the national contact network. These two regions are comparatively isolated in terms of geographic location, with poor accessibility and relatively low social dynamics, which in turn results in fewer inter-city contacts. The emergence of such changes over the study period was influenced by a number of factors, including policy, economics and the environment, but the ongoing impact of the epidemic cannot be ignored [44–46].

In conjunction with the hierarchy of network contacts, it was revealed that Beijing, Guangzhou, Shanghai and Xi'an took the lead in forming a comparatively well-established network of high-level city contacts at the periphery of the center compared to their neighboring cities. In addition, there were two stages, namely the semi-ring-shaped network centered around port cities on the eastern coast and the circular high-level city contact network centered around provincial capitals in each province.

As can be viewed in Table 2, the national density values were above 0.2 after the second quarter of 2021, with a slight decrease in the second quarter of 2022. The density values of urban contact networks in the eastern region were considerably higher than those in the other regions, with frequent population flows and intense connections both inside and outside the cities. The density of in-house networks was significantly lower in the western region than in the other three regions, indicating that cities in the western region were less connected to each other, which is closely related to their natural conditions. In the northeastern region, the network of city connections was "high inside and low outside", with the density of in-house connections being significantly higher than the density of cross-regional mobility networks due to the region's historical experience and humanism, which makes it a very united region. Longitudinally, a downward fluctuation in urban connectivity is observed between the first quarter of 2021 and the second quarter of 2022, corresponding to the development of the epidemic.

### 4.3. Analysis of the Influencing Role of COVID-19 in Urban Contact

4.3.1. Construction of the Indicator System

Population flows constitute one of the significant aspects of urban contact and are influenced by individual will, natural and human–social conditions in the outflow and inflow areas, and government intervention. These factors determine the direction and speed of population flows, as well as the rank of cities in the urban contact network, which in turn shape the structural characteristics of urban contact networks. Resilient urban contact networks tend to remain stable when external shocks are relatively minor. Nevertheless, inter-city contact networks have undergone a process of fractalization, disintegration and remodeling in the face of pandemic shocks [8]. In the course of reshaping, the sensitivity of urban contact networks to the pandemic has equally demonstrated the intensity of urban resilience. Nevertheless, the development status of outflow areas and inflow areas has not only been restricted to the evolution of the development of the pandemic but has also involved variations and differences in other natural and human attributes.

Slope stands out as one of the significant topographic feature factors and plays an essential role in terrain surface analysis [47]. The night-time lighting index mirrors the lighting characteristics of the region and has been adopted by numerous research studies to characterize economic development [48]. In addition, all levels of government in China exert resource dominance, whereby the higher the administrative level of a city, the more political resources it boasts, and this is one of the influencing factors for the tendency of population flows in urban contacts. As a consequence, municipalities directly under central government jurisdiction are assigned a value of 5, national central cities are assigned a value of 4, provincial capitals are assigned a value of 3, cities with separate planning are all assigned a value of 2, and other cities are assigned a value of 1; these values characterize

the political resources of cities [49]. A high-quality ecological environment exerts a tremendous influence on improving the livability of cities, whereas the ecological quality class is classified in accordance with the ecological condition index, which can identify the ecological environment of cities in a scientific and holistic manner.

For this reason, this study constructed an index system, which involved four control variables, namely natural base, economic conditions, political resources and ecological environment, while probing the influencing role of the pandemic in urban contact from the perspective of population flows (Table 3).

**Table 3.** Index system of influencing factors.

| Types of Variables | Main Dimensions | Specific Indices |
| --- | --- | --- |
| Explained Variables | Urban Contact | Urban Centrality |
| Core Variables | Development of Pandemic | Number of Diagnosed Cases |
| | Natural Base | Slope |
| Control Variables | Economic Condition | Night-Time Lighting Index |
| | Political Resource | Urban Political Level |
| | Ecological Environment | Ecological Quality Level |

### 4.3.2. Impact Study on the Basis of National Perspective

With a view to analyzing the evolution of the influencing role of the pandemic in urban contact networks, this study separately probed the overall correlation variation and spatial heterogeneity fluctuation of the two factors from a national perspective while also exploring the evolution of the influencing role at various scales from general to local.

(1)　Overall Correlation Analysis

In this study, the Pearson correlation analysis was employed to calculate the correlation between the two factors at various time periods, a heat map of which was drawn for visualization analysis (Figure 5).

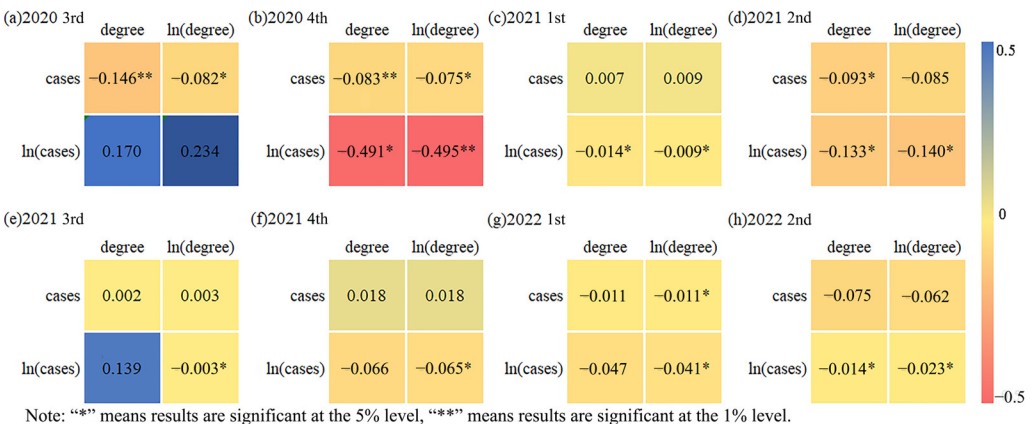

Note: "*" means results are significant at the 5% level, "**" means results are significant at the 1% level.

**Figure 5.** Heat map of Pearson correlation coefficient between urban contact networks and epidemic development.

According to the results of the significance test in the overall scale analysis, it was found that COVID-19 exerted a negative effect on urban contacts, where the prevalence of the pandemic resulted in a temporary slowdown of economic development and social vitality in the area, as well as restriction of both internal travel and cross-city population flows, thereby impeding urban contacts. Nevertheless, the color of the heat map becomes considerably lighter after 2021, indicating that the influence of the pandemic became increasingly diminished and insignificant, while the sensitivity of urban contacts to the pandemic progressively decreased. This result also verifies the correctness and foresight of the decision to "designate high-risk areas in general on the basis of units and buildings",

which in turn confirms the rationality of the optimization of the management and control of the pandemic toward the direction of openness. Despite the tendency of this impact to increase in April 2022, this was closely related to the emergence of new mutant strains of COVID-19 with minor fluctuations.

(2)  Spatial Heterogeneity Analysis

The influencing role of the development of the pandemic in urban contacts requires attention to be given to spatial differences. This study took into account the difference in urban spacing as a smoothing treatment of the differences in the search radius, with local characteristics of the influencing role explored on the basis of MGWR 2.2 software using the MGWR method (Figure 6).

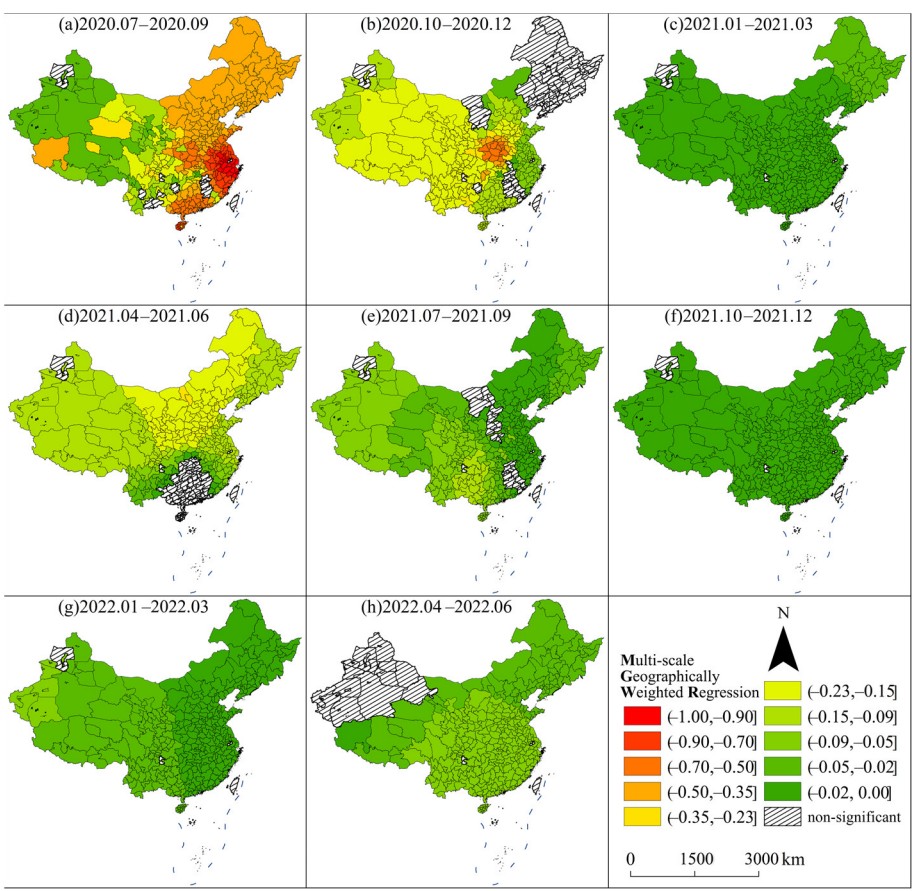

**Figure 6.** Spatial distribution of the regression coefficients of urban contact networks in the development of the pandemic from a national perspective.

According to the local analysis, it is apparent that the changing trend of the influencing role of the pandemic in urban contact from the spatial perspective of population flows is consistent with the holistic perspective. At the very beginning of the study period, the influencing role of the pandemic remained relatively strong, with the eastern region and the northeastern region being more sensitive to it while being the fastest recovering regions; the central region was less resilient, while the negative influencing role of the pandemic weakened markedly to near zero from 2021 onward. In the second quarter of 2021 and the second quarter of 2022, there was a higher upward trend in the central region as a result of the emergence of a variant strain of COVID-19. It has been demonstrated in studies that the infectivity of Omicron was about 37.5% higher than that of Delta. Nevertheless, the increase in the negative effect of the Omicron variant strain from a local perspective was noticeably lower than that of the Delta variant strain in April 2021, which in turn indicated that the shock effect of the pandemic had been dramatically diminished

with the resilient rebound effect of the urban contact network, resulting in a more robust capacity of cities to cope with major public health events.

### 4.3.3. Impact Study on the Basis of Regional Perspective

In accordance with the socioeconomic development status, China is categorized into four economic regions, namely the eastern region, central region, western region and northeastern region, which scientifically mirror the socioeconomic development status of various regions in China. The population flows within each region make up the internal urban contact network, while the population of other regions flowing into one region and the population of one region flowing out to other regions make up the cross-regional urban contact network from the perspective of "population inflow" and "population outflow" of this region, respectively. Meanwhile, the local regression coefficients of various networks and the development of the pandemic were sequentially probed, with the distribution map of MGWR coefficients being plotted (as shown in Figures 7–9). The influencing role of the pandemic in urban contact networks requires an emphasis on the sensitivity and vulnerability of various contact network types and regions to the pandemic.

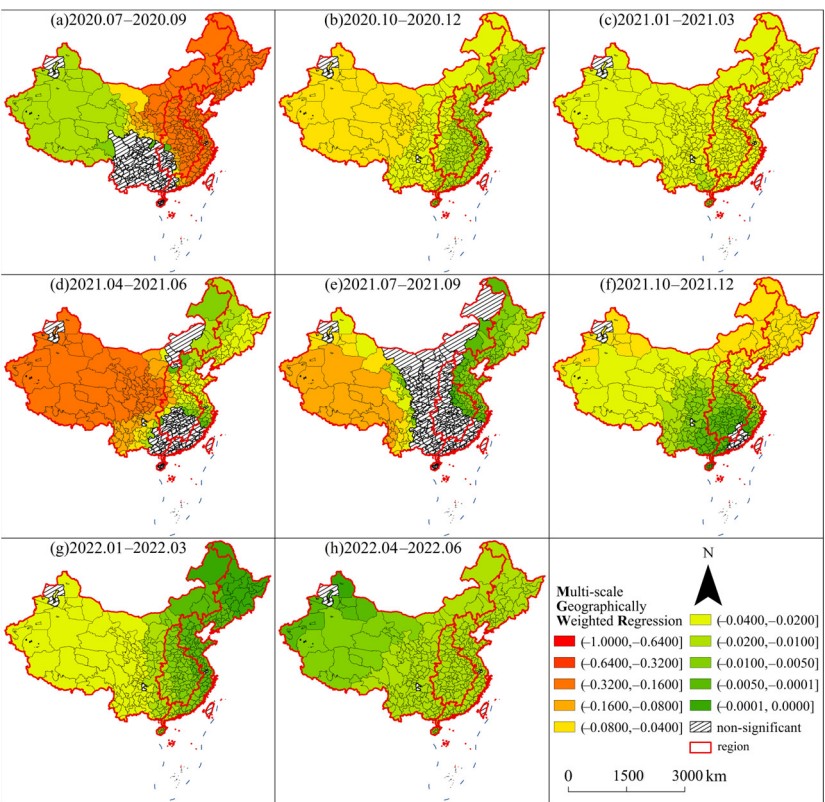

**Figure 7.** Spatial distribution of the regression coefficients of urban contact networks in the development of the pandemic within each region from a regional perspective.

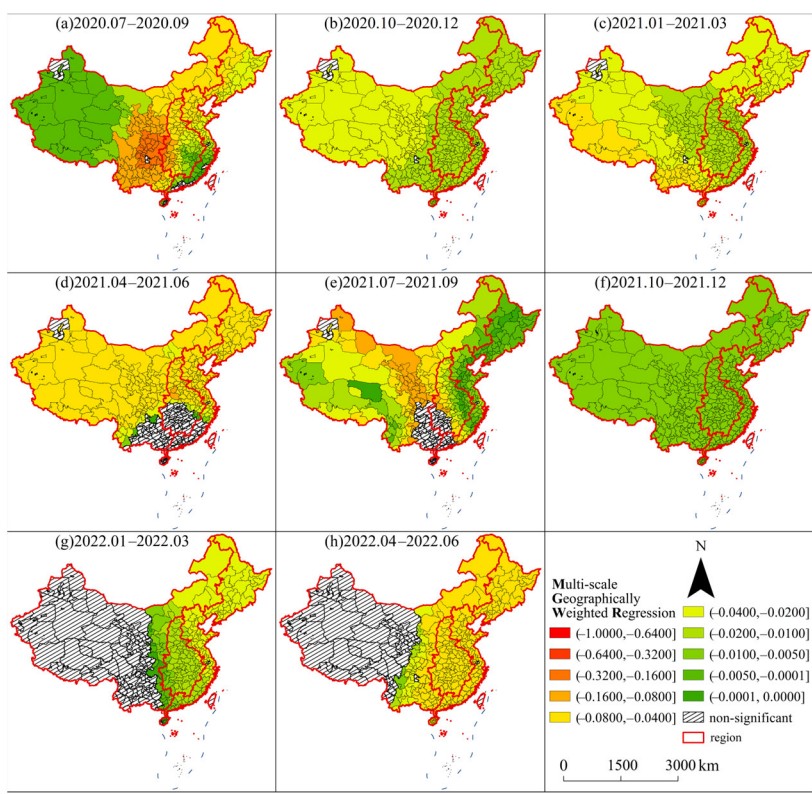

**Figure 8.** Spatial distribution of the regression coefficients of urban contact networks in the development of the pandemic from the perspective of regional population inflow.

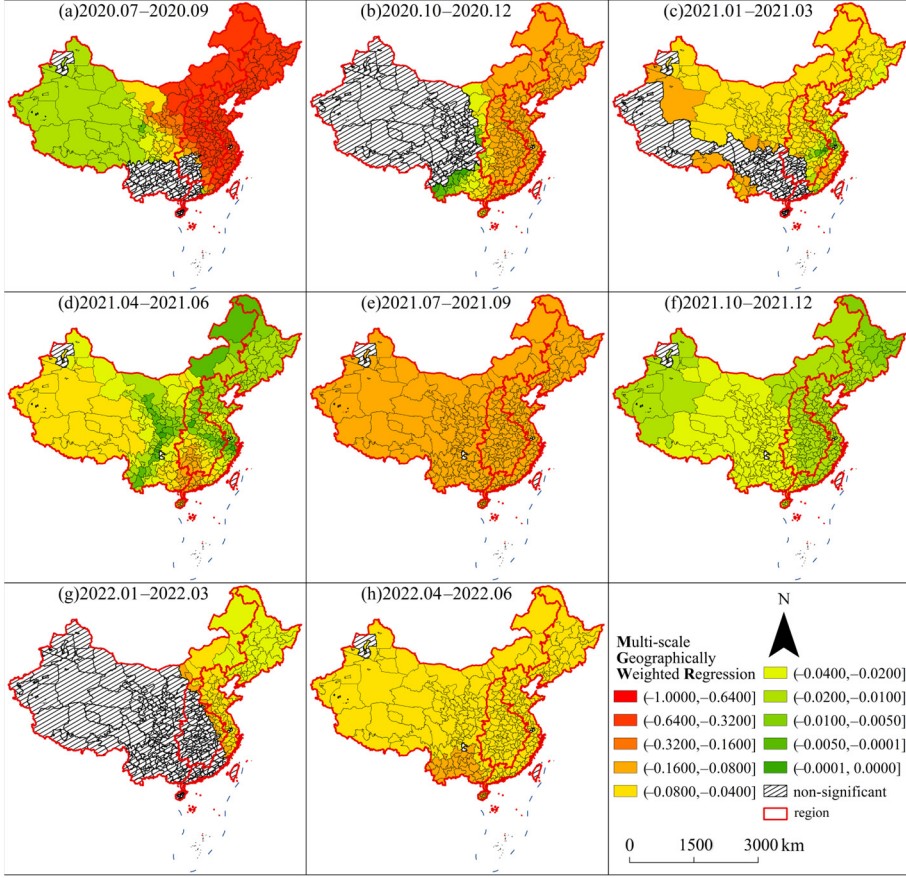

**Figure 9.** Spatial distribution of the regression coefficients of urban contact networks in the development of the pandemic from the perspective of regional population outflow.

As can be gleaned from Figure 7, the intra-regional urban contact network was initially most affected by the pandemic in the eastern region and the northeastern region, followed by the central region, and least affected in the western region. Nevertheless, following the emergence of the mutated strain of the Delta virus in China, the vulnerability of the regional urban contact network was reversed, with the western region being the most affected, while the intra-regional contact network in the eastern coastal region was more robust and resilient in view of the scientific nature of the preventive measures, the high-end medical conditions and the effective management and control strategies.

In conjunction with the distribution of the regression coefficients of cross-regional urban contact networks from the perspectives of "population inflow" and "population outflow", the study revealed that the regional boundaries of the four major economic regions in China exhibit certain boundary effects, whereby the influencing role of urban contact networks formed by cross-regional population flows and intra-regional population flows differs substantially. By synthesizing the distribution map of the regression coefficients of the two urban contact networks formed by inflows and outflows, the study revealed that the sensitivity of the central–western junction exerts special effect on intercity contact networks, which span longer distances across the region. The powerful second-tier cities in China, such as Chengdu, Zhengzhou, Changsha, Wuhan, Xi'an and Chongqing, are all located in this region, where tourism serves as a key driver of recent development, with a significant capacity to absorb mobile populations. In addition, the development of the pandemic in these regions will lead to the cancellation of the movement of numerous populations that would otherwise flow into the region, while the contact network constituted by outflowing populations will not be affected.

It is notable that the negative effect of the pandemic is commonly higher for urban contact networks composed of outflow populations than for other networks. This demonstrates that the willingness of individuals themselves to move over long distances diminishes considerably in the face of the pandemic, even if they have already entered the pandemic normalization phase, where the emergence of mutant strains of the virus has a significant impact on long-distance ties. Among them, the Yangtze River basin, the Yellow River basin and the northeastern region, where the urban agglomerations are located, exhibit stronger network resilience and intense inter-city ties, which are capable of breaking through the border barrier effect to a certain extent.

## 5. Discussion

### 5.1. Complexity of the Relationship between the Development of Pandemic and Urban Contact

At the inception of the pandemic, numerous studies have identified a substantial relationship between the pandemic, population flows and urban contact, while the majority have concentrated on the role of human behavior in the spread of the pandemic [50,51]. Nevertheless, the interactions between the pandemic and urban contact do not emerge in a direct manner, nor are they straightforward single-directional effects; rather, there exist multiple indirect bi-directional effects of information shocks, individual intentions and government interventions [52,53]. As a consequence, research should not be restricted to probing the unidirectional driving effect of human activities on the pandemic but should also take into account the reverse inhibitory effect of the pandemic. In conjunction with the findings of the study, it is apparent that the pandemic exerts a significant negative effect on urban contact, which in turn verifies the existence of a bi-directional effect.

The study revealed that the pandemic still exerted a complicated negative effect on urban contact from a spatial perspective of population flows during the study period, which is consistent with previous studies [54,55]. Nevertheless, this study demonstrated that this negative effect diminished progressively in both overall and local contexts, which substantiates the efficiency, science and foresight of the anti-epidemic policy in China. As a consequence, a low mortality rate and minimal economic disruption were achieved, resulting in a credible and applicable "Chinese solution to harmonize the prevention and

control of the pandemic with socioeconomic development". In addition, the research results also illustrated that despite the phased success of the prevention and control of the pandemic, the study shared some of the same results focusing on spatial heterogeneity [7,56,57]. The study revealed that the susceptibility of different cities to the pandemic differed substantially, while there also existed differences in the results exhibited by cities at various network scales and types of mobility networks. The coastal cities in the eastern region are economically developed with easy access to external transportation, which makes them most vulnerable to the influx of pathogens, thereby being impacted by the pandemic. Nevertheless, this region was the fastest to recover after entering normalized pandemic preparedness, as well as being the most sustainable after confronting the invasion of mutant strains of viruses, such as Delta and Omicron. Despite its high altitude, slope, complicated terrain and poor transportation conditions, the western region is less likely to be exposed to pathogens during the pandemic; however, the region suffers from poor medical conditions, weak medical treatment capacity and a deficiency in the resilience of urban contact networks. Cities in the central region are comparatively weak in internal contacts, while the network resilience is comparable to that of the eastern region, with a stronger capacity to alleviate the shocks of a pandemic. The urban contact network in the northeastern region features distinctive characteristics, with tight internal and sparse external connections, while the network is resilient and less sensitive to the pandemic. In this regard, it can be concluded that the impact of the pandemic on urban contacts from the spatial perspective of population flows is complicated, with mechanisms that can be further elaborated. At the present time, the prevention and control policy for the pandemic in China has been progressively liberalized from containment and control. This optimization is not merely a liberalization but a progressive restoration of social order while taking into account the spatial differences in the sensitivity of urban contact to the development of the pandemic in the process of that optimization. As a result, it has contributed to the economic recovery, social prosperity and normalization of people's lives.

### 5.2. Variations in the Influencing Role of the Mutant Strain of the Virus after Its Emergence

In comparing the study hypotheses, the results were in accordance with the initial hypotheses, with the exception of some peculiarities. During the study period, COVID-19 mutated repeatedly, including a highly infectious mutant strain from Delta to Omicron [58]. The resilience of the urban contact network was continuously examined through the further impact triggered by mutant strains. At the present time, COVID-19 continues to mutate; despite studies that have identified a gradual erosion of virulence, the sensitivity of urban contacts may fluctuate in response to the emergence of mutant strains, making strategies and policies to cope with the emergence of mutant strains highly relevant for urban development. The results demonstrated that the prevalence of Delta and Omicron exerted certain hindering effects on urban contact, with Omicron exhibiting greater infectivity, spread and fugacity than Delta, whereas the hindering effects induced by Omicron prevalence were minor compared to the effects of the Delta virus. As a consequence, it is apparent that the efficiency of the improvement in the capacity of Chinese cities to cope with major public health events is higher than the rate of improvement in infectiousness by virus variation.

### 5.3. Impact of Control Variables on Urban Contacts from a "Population Flows" Perspective

This study acknowledged that urban contacts were simultaneously impacted or disturbed by multiple factors. For the sake of independently clarifying the impact brought about by the development of COVID-19 on urban contacts, as well as preventing the potential issues of disregarding the impact of other factors and amplifying the impact of independent variables, this study set control variables of the natural base, economic conditions, political resources and ecological environment to construct an index system. Over the course of the study, it was identified that the night-time lighting index, urban political

level and ecological quality level exert a positive effect on urban contacts, which in turn advance improvement in the levels of cities in the urban contact network, where the urban political level exerts the strongest driving effect. On the contrary, the higher the slope value, the lower the urban centrality and the weaker the urban contact. In accordance with the availability of data and the empirical results of the research model, it is evident that the control variables in the index system were selected in a scientific and effective manner, which avoided the endogeneity between the core variables and the dependent variables to a greater extent.

*5.4. Limitations*

In this study, big data on population migration from the Baidu Migration Platform were employed to calculate urban contacts from the perspective of population flows, with this data source featuring the advantages of massive samples, high real time and high coverage. As a consequence, in comparison with the available traditional research results on population flows, this study broke through the inter-provincial level regarding the research unit while making an attempt to examine the inter-city population migration paths and improve the issue of non-directional and non-weighted urban contact networks found in most studies. Nevertheless, the above analysis is based on the assumption that internet users represent a large proportion of the total resident population in the study area and that the structural proportions are consistent with information distribution preferences. In the case of China, with a mobile phone subscriber penetration rate of 119.2 units per 100 people and an internet penetration rate of 74.4% as of June 2022, the scale of internet users makes the study results more reliable, but the extent to which this assumption will introduce bias when the research mechanism framework is applied in other international systems in the future is something that needs to be explored further. A comprehensive multi-dimensional measure of urban connectivity is yet to be developed to test the magnitude of the research error and to balance the biased nature of the data resulting from this hypothesis.

This study places emphasis on probing the spatial and temporal characteristics and evolutionary trends of the impact of the pandemic at multiple scales, where the results of local analysis based on regional scales indicate that there exists a consistency in the impact of urban contact networks on the development of the pandemic within most urban agglomerations in China. As a consequence, the impact of the pandemic on urban contact networks at various urban agglomerations scales is worthy of further discussion.

## 6. Conclusions

On the basis of daily new cumulative diagnosed data and population flow data on COVID-19 from 1 July 2020 to 30 June 2022 in 366 cities across China, this study employed SNA, the Pearson correlation analysis and the MGWR to determine the impact of COVID-19 on urban contact networks in China over a longer period of time during the outbreak of the pandemic from a population flow perspective. This resulted in the following conclusions:

(1)  From a national perspective, the number of diagnosed cases in China was relatively low during most of the study period, with increases occurring solely during the shorter period of time when mutated strains of the virus appeared. From a regional perspective, the western region was compatible with the overall situation in China, the central region exhibited an increase in the number of cases attributable to mutated strains, while the eastern region and the northeastern region exhibited broadly analogous and more stabilized trends. The population flow perspective of inter-city contacts increased the density of urban contact networks over time. In conjunction with the hierarchy of network contacts, it was identified that Beijing, Guangzhou, Shanghai and Xi'an with their neighboring cities took the lead in developing a more sophisticated center–periphery network of high-grade urban contacts.

(2) The effect of COVID-19 on urban linkages was consistent with Hypothesis 1. From a national perspective, COVID-19 had an overall negative relationship with urban linkages in China, with the pandemic impeding urban linkages. As the epidemic progressed, the effect of COVID-19 on urban linkages became weaker and less significant, while local studies verified the existence of Hypothesis 2, reflecting significant spatial heterogeneity in the effect, with a clear weakening effect in the eastern region ≈ northeastern region > central region > western region.

(3) From a regional perspective, the intra-regional urban contact network was initially shaken most by the pandemic in the eastern region and the northeastern region, followed by the central region, and least in the western region. The intra-regional urban contact network in the eastern region and the northeastern region recovered swiftly, while the resilience of the eastern coastal region manifested in a more scientific regional contact network. The difference in sensitivity to the pandemic between networks formed by cross-regional and intra-regional population flows is remarkable. By integrating the dual perspectives of "population inflow" and "population outflow", the regional boundaries of the four major economic regions in China exhibit border effects, with the outflowing population constituting an urban contact network, which typically exerts a higher negative effect on the pandemic than other networks.

In a nutshell, the pandemic maintains a negative effect on urban contact, but this effect has progressively diminished. The shock effect of the pandemic has been dramatically reduced by the resilient rebound of the urban contact network, while the capability of Chinese cities to cope with major public health events has become increasingly robust. Furthermore, this study also demonstrated that the pandemic prevention policies in China have been reliable, appropriate and imperative, with adequate recognition of the science and forward planning of the progressive management and control of the pandemic.

**Author Contributions:** Conceptualization, Xuejie Zhang and Jinli Zhao; methodology, Xuejie Zhang and Jinli Zhao; software, Xuejie Zhang and Haimeng Liu; validation, Mengcheng Li and Yi Miao; formal analysis, Xuejie Zhang; data curation, Chengxin Wang and Haimeng Liu; writing—original draft preparation, Xuejie Zhang; writing—review and editing, Xuejie Zhang, Jinli Zhao, Yi Miao and Mengcheng Li; supervision, Yi Miao; funding acquisition, Chengxin Wang. All authors have read and agreed to the published version of the manuscript.

**Funding:** This research was funded by the National Social Science Foundation of China (Grant No. 20BJY070) and the Major Project of Key Research Bases for Humanities and Social Sciences Funded by the Ministry of Education of China (Grant No. 22JJD790015).

**Institutional Review Board Statement:** Not applicable.

**Informed Consent Statement:** Not applicable.

**Data Availability Statement:** The data presented in this study are available on request from the corresponding author. The data are not publicly available because research is ongoing.

**Conflicts of Interest:** The authors declare no conflicts of interest.

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
