# Peer review of "A Study of the Impact of COVID-19 on Urban Contact Networks in China Based on Population Flows"

_ijgi, doi:10.3390/ijgi12060252_

Round 1
Reviewer 1 Report
The abstract is unclear. It should be clear enough to a moderately informed audience to allow readers to discern what it is about - I can barely make sense of it.
Around line 83 - some studies - what studies?
Lines 85-86 - Why wouldn't they be? I don't know that this 'gap' has been introduced? Remainder of paragraph - not entirely convinced - could be more effectively stated.
Line 114 - not sure what this means?
At line 116 - spread of information - goes without saying, does it not?
"small-scale silence" - not sure what this means?
Figure 2 - seems limited in explanatory help
around line 155 - sensitivity of urban contacts to the development of the pandemic - could be clearer throughout the discussion how this is being measured or what proxy you are using
line 159 - also around line 174, 'study - about that study - tell the reader more about it - how you obtained data - etc. needs to be clear enough to replicate the work here. where are these data, if they are not yours (i.e. government's data? are they publicly available?)
190, 194 - OK, so we are getting more info about the database, I'm guessing (?) but this is exactly backward - you should be up front with this information - it should not be presented gradually - start the section with the source of it (if it is, and I'm not sure - so you need to clarify).
Lines 210-224 - I don't think you intended to include this, since it is template boilerplate language.
Figure 4 - what are we to make of this?
Line 325 - "cities with high centrality nationwide are predominantly located in the eastern and central regions" and so forth - is this not the direct result of policy and restrictions, or is it due to something else?
351-353 - not clear what you intend.
585-588 - very much concerned about limitations on these data - not sure that this entirely captures the limitations, even if the study may be a contribution.
602- do you have concerns about the validity of case diagnoses (do they track true extent of illness?) and if not, why not? Would this not create some issues with data within the system, and validity of this study?
635- cities being able to cope has become increasingly robust- but there have been issues with lockdowns and resisting such impositions, even to November 2022. What did lockdowns have to do with outcomes seen, if they had any role at all?
Reviewer 2 Report
I did not find a clearly formulated purpose of the research study in the Abstract. The authors wrote what they did, how many cities they took into account and gave other details, but in my opinion it would be good to give a clearly formulated goal of these activities. The same is true of the main body of the article; here, too, I did not find a clearly formulated research goal, but only information about the research tools involved, statistical tools, and the research sample. It would be worth including the sentence: The purpose of the research study was … Regardless of the exact formulation of the general purpose of the research / research study, it would also be worth writing in the content of the article what the scientific (cognitive) and utilitarian (useful) purpose of the research study was. Therefore, I also propose to write: "The scientific aim of the study was ..." and "The utilitarian aim of the study was ...". In addition, I propose to formulate a research problem that was solved by the authors. The formulation of the research problem can be based on several questions that the authors provided in the paragraph, lines 83-88. In the summary of the research problem, a gap in the current state of knowledge can be indicated, which the authors try to fill through their research. In my opinion, the considerations presented by the authors in the Introduction allow both to formulate the research problem and to indicate the gap in the research area under consideration. It is only necessary to call them a research problem and a gap in the article. A natural consequence of this part of the considerations is the formulation of research hypotheses later in the article, which should be considered correct and fully justified.
I do not understand the summary of the data describing in Figure 3. The years (2020, …) are given above the graphs, while the numbers are given below the abscissas (x). Are these months that have been generally called "time"? Readers, like me, will have to guess what the numbers in the charts mean, and in my opinion all descriptive information should be given in an unambiguous way. Therefore, I think that the graphs in Figure 3 require a more detailed description in order to easily interpret the intentions of the authors regarding the presented information.
If the acronym MGWR is given in the legend of Figures 6-9, it would be good to explain what this acronym means in the figure caption. Of course, there is a full explanation of the model acronym in the text of the article, but it would be nice if this information was directly available when analyzing the Figures.
In the analysis of research results / discussion, it would be worth developing the issue of the possibility of applying the presented research approach in other regions of the continent / world. Do the presented model solutions also take into account the exchange / movement of population in the international system? In my opinion, this aspect is also important from the point of view of the number of COVID-19 cases in the country and can be developed in the context of further research / analysis.
In the Conclusions chapter, it would be worthwhile for the Authors to refer in detail to Hypothesis 1 and Hypothesis 2, because such two hypotheses were put forward in the chapter "Theoretical Analysis and Research Hypothesis". Only in the conclusion (2) the authors refer to the hypothesis, but it is not known which one, the first or the second. Therefore, I suggest verifying the Conclusions in such a way as to explicitly refer to Hypothesis 1 and Hypothesis 2.
Round 2
Reviewer 1 Report
The authors have responded acceptably to my previous feedback, and significantly improved the paper in this most recent draft. There has been clear effort to respond to the comments given.